# Effects of Moisture, Temperature, and Salt Content on the Dielectric Properties of Pecan Kernels during Microwave and Radio Frequency Drying Processes

**DOI:** 10.3390/foods8090385

**Published:** 2019-09-02

**Authors:** Jigang Zhang, Maoye Li, Jianghua Cheng, Jiao Wang, Zhien Ding, Xiaolong Yuan, Sumei Zhou, Xinmin Liu

**Affiliations:** 1Tobacco Research Institute, Chinese Academy of Agricultural Sciences (CAAS), Qingdao 266101, China; 2School of Plant Protection, Anhui Agricultural University, Hefei 230036, China; 3Institute of Food Science and Technology, Chinese Academy of Agricultural Sciences (CAAS), Beijing 100193, China

**Keywords:** pecan, dielectric properties, radio frequency (RF) heating, simulation

## Abstract

Dielectric properties of materials influence the interaction of electromagnetic fields with and are therefore important in designing effective dielectric heating processes. We investigated the dielectric properties (DPs) of pecan kernels between 10 and 3000 MHz using a Novocontrol broadband dielectric spectrometer in a temperature range of 5–65 °C and a moisture content range of 10–30% wet basis (wb) at three salt levels. The dielectric constant (ε′) and loss factor (ε″) of the pecan kernels decreased significantly with increasing frequency in the radio frequency (RF) band, but gradually in the measured microwave (MW) band. The moisture content and temperature increase greatly contributed to the increase in the ε′ and ε″ of samples, and ε″ increased sharply with increasing salt strength. Quadratic polynomial models were established to simulate DPs as functions of temperature and moisture content at four frequencies (27, 40, 915, and 2450 MHz), with *R^2^* > 0.94. The average penetration depth of pecan kernels in the RF band was greater than that in the MW band (238.17 ± 21.78 cm vs. 15.23 ± 7.36 cm; *p* < 0.01). Based on the measured DP data, the simulated and experimental temperature-time histories of pecan kernels at five moisture contents were compared within the 5 min RF heating period.

## 1. Introduction

Pecan [*Carya illinoinensis* (Wangenh.) K. Koch] is a world-famous Juglandaceae tree nut, mainly distributed in North America, including the United States and Mexico, which accounts for approximately 60% and 30%, respectively, of global pecan nut production [1]. The pecan nut has gained increasing popularity due to its abundant nutrient components (high unsaturated fatty acids, protein, minerals, vitamins, phenolics, flavonoids, phytosterols, and saponins), unique buttery flavor, and potential health-promoting benefits, such as modulating blood cholesterol levels, preventing coronary heart disease, and mitigating adiposity [2,3,4,5,6]. Suitable postharvest drying is an essential step in maintaining the quality and active ingredients of pecan nuts [7].

The moisture content of freshly harvested pecan nuts must be promptly reduced from 25–35% wet basis (wb) to less than 6% wb through dehydration to decrease the metabolic rate, which facilitates the subsequent transport, storage, and processing of pecan nuts [8,9]. Pecan kernels are enclosed in a thick hard shell, which leads to a slow drying cycle. Often, more than three days is required to accomplish dehydration through sun or air convection drying methods [10]. Moreover, these traditional convection drying methods require a large area to lay the pecans outdoors and are susceptible to ambient climatic circumstances, which often cause undesirable quality degradation, such as mold, being off-flavor, or discoloration [11]. Therefore, advanced drying technologies have become an urgent issue that needs to be resolved for postharvest pecan processing.

Dielectric heating is a fourth-generation heating treatment technology that is widely used for the industrial drying of food, agricultural products, and industrial materials [12,13]. When dielectric samples are subjected to a rapid reversal electric field, the electromagnetic wave passes through the material shell and interacts directly with the permanent or induced dipoles and charges inside the sample to induce underlying dipole polarization and ionic conduction with finite displacement, resulting in heat generation inside the volume [14,15]. However, to design and control the dielectric heating process, knowledge regarding the dielectric properties (DPs) of materials is essential, which can be used to determine how much polarization of dielectrics and charge conduction can occur and dissipate when subjected to an electromagnetic field [16]. The DPs of materials are normally described using the complex relative permittivity, ε*, which is represented as follows: ε* = ε′ – jε″. The real part (ε′) of the formula is named the dielectric constant, which reflects the charge storing capability of the material regardless of the sample’s size. The imaginary part (ε″) is named the loss factor, reflecting the energy dissipation in the material due to the conversion of the electromagnetic field into heat energy. The higher the dielectric loss factor values, the higher the electromagnetic energy that is absorbed and converted by the material, and the higher the rate of the temperature increase [17]. Therefore, DPs are the most basic parameters for characterizing the interaction, thermal efficiency, and penetration depth when designing MW and RF heating processes [18].

Several studies have reported the DPs of foodstuffs over various frequency, temperature, moisture content, and salinity ranges in drying, pasteurization, and pest control [19,20,21]. The DPs of macadamia kernels were determined in the frequency band of 10–1,800 MHz within a temperature range of 25–100 °C at moisture contents of 3–24% wb by adopting open-ended coaxial probe technology [22]. Zhang et al. [23] found that the DP value of peanut kernels decreased with increasing frequency but increased with increasing temperature and moisture content. Ling et al. [24] simulated quadratic polynomial equations for the temperature, moisture content, and frequency of non-salted pistachio nuts, according to the DP data measured from 10–4500 MHz at 25 to 85 °C. Jeong et al. [25] found that RF heating can potentially inactivate foodborne *Salmonella enterica* in pistachios and that the inhibitory effect is controlled by the dielectric loss factor relative to the salt content. The dielectric heating of pecans can effectively prevent the attack of weevil [26], and the dielectric heating does not make the color of the epidermis darker when stored later than steam [27]. In recent years, new technologies of broadband dielectric spectroscopy have been developed to measure the DPs of materials. These new technologies have integrated systems, are easy to operate, have a broadband frequency range, and provide more accurate results in comparison to the previous method of the open-ended coaxial-line probe. However, some problems exist in the direct application of a broadband dielectric spectrometer for determining the DPs of irregularly shaped materials because this method requires close contact between the samples and parallel electrode probes during measurements. The contact problem is overcome by creating compressed cylindrical samples with flat surfaces from a ground sample to match the kernel bulk density of the pecan kernel. 

In this study, the objectives were to determine the DPs of pecan nut kernels over frequencies from 10 to 3000 MHz at moisture contents between 10 to 30% at four temperature levels. Furthermore, simulations were performed based on empirical equations describing the DPs of the pecan kernels as functions of the moisture content and temperature at certain frequencies. The effect of the salt strength (mild, medium, and heavy) on the DPs of pecan nut kernels was also assessed. The penetration depth of electromagnetic energy into the pecan kernels under these different conditions was determined. Engineering insights into the implications of these DPs for the RF process during 5 min of RF heating for pecan samples with five different moisture contents were discussed.

## 2. Materials and Methods

### 2.1. Materials

Pecan fruits of the Mahan variety were harvested from a local farm in Jiande (Jiangsu Province, China) in September 2016. Ethephon aqueous solution (4000 mL kg^−1^) was sprayed on the surfaces of the fruit hulls to accelerate the hull separation from the shell. The in-shell pecan nuts were immersed in sterilizing liquid (NaClO, 5 mg mL^−1^) for 5 min and then rinsed with deionized sterile water. The surface of disinfected nuts was air-dried for further grading treatment. High-quality samples free of any defects were selected based on full and plump size and similar sense and stored at 4 °C until further testing. The main nutritional ingredients of the pecan samples were measured with Association of Official Analytical Chemists (AOAC) standard methods (Table 1).

### 2.2. Preparation of Pecan Kernel Samples

Pecan nut kernels (200 g) with an initial moisture content of 30.2% wb were divided evenly into four sublots and then dried to moisture content levels of 25, 20, 15, and 10% wb at 40 °C in a blast drying oven. The samples with different moisture contents were then vacuum sealed in polyethylene bags at 4 °C until further use. The storage time was assumed to be sufficient for the residual humidity in each kernel to completely relax and reach a uniform level throughout the kernel.

To quantitatively examine the effect of the salt content on the DPs of the pecan kernels, the salted pecan samples were prepared in accordance with the method of Ling et al. [24]. A total of 150 g of fresh pecan kernels was freeze-dried to adjust the moisture content to 5%. The dried nut kernels were then divided into three equal parts that were immersed in 1000 mL of brine (NaCl 5%, 10%, and 20% w v^−1^) for 60 min with constant stirring to simulate the market products named lightly-, medium-, and heavily-salted nuts. These soaked salty samples were removed and freeze-dried again to obtain an end moisture content of 15%. The dried pecan samples contained the same ingredients and had three levels of salt: Light, medium, and heavy. All the samples were stored (relaxing time was sufficient) for further use at 4 °C in individual polyethylene bags.

### 2.3. Preparation of Cylindrical Samples

To obtain close contact between the samples and gold-plated parallel electrodes during the measurement of DPs, cylindrical samples whose density closely matched the actual kernel density were prepared. The prepared kernels with different moisture and salt contents were ground into powder. The powdered samples were then compressed under vacuum in a metal cylindrical mold of a hydraulic press (YP-2, Shanghai Shanyue Science Instrument Co., Ltd., Shanghai, China) to form cylindrical discs (ø: 7 mm; H: 3 mm). To control the pressure of the hydraulic press and the weights of the added powder, the density of the compressed cylindrical sheets closely matched the density of the real kernel sample.

The kernel bulk density of the pecan kernel samples with different moisture contents was determined using the liquid displacement method [24], in which toluene (C_7_H_8_) was used as the displaced liquid due to its low surface tension and non-absorption by nut kernels. The actual density was distributed within the numerical zone from 0.8150 to 1.2715 g cm^−3^ (Table 2) as the moisture content varied between 10% and 30%. The specific heat at different densities was determined by a dual-needle probe method [28] using a thermal properties analyzer (KD2 Pro, Decagon Devices, Pullman, WA, USA).

### 2.4. Measurement of DPs

The DPs of the cylindrical sheet samples were measured using a Novocontrol broadband dielectric spectrometer system (Novocontrol Concept 80, Montabaur, Germany). The key components of the system comprised an Alpha-A dielectric analyzer, Quatro temperature control systems, a Novocontrol BDS 1200 sample cell, a Dewar liquid nitrogen system, a computer, WinDETA data acquisition, and evaluation software. In this research, the DPs of samples with different moisture and salt contents were determined in the frequency band of 10 to 3000 MHz at four temperatures (5, 25, 45, and 65 °C). The temperature was accurately regulated by the Quatro temperature control systems coupled to the dielectric analyzer. Prior to measurement, the computer was switched on, as was the dielectric analyzer to keep it in a stable state. The cylindrical sheets were then sandwiched between the parallel electrodes and installed in the sample cell that could be immersed in a nitrogen environment to condition the tested samples to the set temperatures prior to each detection. The samples were first gradient frozen (10 °C min^−1^) to 5 °C, the equilibrium was maintained at this temperature for 3 min, and then the DPs were measured at intervals of 20 °C. Approximately 20 min was required for the temperature of the tested sample to increase from one level to the next.

### 2.5. Power Penetration Depth

The penetration depth (dp) refers to the quantitatively determined effective acting distance (in meters) between the MW or RF power and materials, where an incoming power intensity is decreased to 1/e (e = 2.7182) of its amplitude transiting the surface. The dp can be calculated using Equation (1):(1)$$\mathrm{dp}=\frac{c}{2\pi f\sqrt{2\varepsilon^{\prime }\left[\sqrt{1+{\left(\frac{\varepsilon^{\prime \prime }}{\varepsilon^{\prime }}\right)}^{2}}-1\right]}}$$
where c represents the speed of light in a vacuum (3 × 10^8^ m s^−1^), f represents the frequency (Hz), and ε′ and ε″ are the permittivity and loss factor, respectively. The dp of the pecan samples was calculated according to the measured dielectric data at optional frequencies, temperatures, moisture contents, and salt contents.

### 2.6. RF Heating Process

Further research is needed to provide engineering insights into the implications of these DPs for the RF heating process, and we are seeking to explore the relationship between DPs and the RF heating process in which the heating rate of pecan is investigated as the consequence of those dielectric properties for an RF heating process, using both experiment and simulation. A 6 kW, 27.12 MHz parallel plate RF system (SO6B, Strayfield International, Wokingham, UK) was used in this research. The RF system includes a pair of parallel electrodes, controller, and RF cavity. The RF power of the system can be changed by adjusting the electrode gap between the upper and lower plates (90–190 mm) to achieve different heating rates for the material. An electrode gap of 150 mm was selected during 5 min RF heating. The pecan samples (3 kg) with different moisture levels (10%, 15%, 20%, 25%, and 30% wb) were successively placed in the polyethylene container on top of the bottom (ground) electrode for dielectric heating, where the stacked height of the sample should be less than the dp. The temperature increase at the central position of the heated sample was monitored using a fiber optical temperature sensor system (HQ-FTS-D1F00, Heqi Technologies Inc., Xian, China).

When the sample is heated in the RF cavity, the RF electric field acts as a heat source and leads to heat transfer inside the material. The heat transfer equation during RF heating can be calculated by Equations (2) and (3):(2)$$\frac{\partial T}{\partial t}=\frac{2\pi \cdot f\cdot \varepsilon_{0}\cdot \varepsilon^{\prime \prime }\cdot |\vec{E_{m}}{|}^{2}}{\rho \cdot C_{P}}$$
(3)$$|\vec{E_{m}}|=\frac{V}{\sqrt[{2}]{{\left(\varepsilon^{\prime }d_{0}+d_{p}\right)}^{2}+{\left(\varepsilon^{\prime \prime }d_{0}\right)}^{2}}}$$
where T is the temperature increase in the material (°C); t is the temperature rise time (s); f is the frequency (Hz); ε_0_ is the the dielectric constant of free space (8.854×10^−12^ Fm^−1^); E is the electric field intensity (Vm^−1^); Cp is the specific heat of material (J·kg^−1^C^−1^); ρ is the density of the material (kg m^−3^); V is the voltage between the electrodes; d_0_ is the air gap from the top electrode plate to the upper surface of samples; and dp is the height of the sample. Equation (2) shows that T is proportional to the material’s ε″ and can be calculated by the measured ε″ data. This increase in temperature is theoretically attributed to dielectric heating in comparison with the experimentally measured increase in RF heating.

A computer simulation model for solving coupled electromagnetic and heat transfer equations based on the RF system was constructed using COMSOL software (V4.3a, COMSOL Multiphysics, CnTech Co., Ltd., Wuhan, China). The modeling steps include creating FEMLAB (AC/DC) modules, a geometrical model of RF systems, a heat transfer module, assigning initial and boundary conditions, mesh creation and optimization, choosing solver, setting tolerance, and time steps, and solving inbuilt convergence [29]. The measured DP value of pecan kernels was put into the heat transfer module to solve the coupled heat transfer equations, and simulated temperature rise values of samples were saved. The upper electrode voltage had a constant value of 6000 V with ±5% fluctuation. The mesh system included 132,279 domain elements (tetrahedral), 10,874 boundary elements (triangular), 843 edge elements (linear), and 26 vertex elements. The direct linear system solver (UMFPACK) was used with a relative tolerance and absolute tolerance of 0.01 and 0.001, respectively, with the initial and maximum time steps of 0.001 s and 0.1 s. These computer simulations were performed by a Dell workstation (Dell, Inc, Texas, USA) with 8 GB RAM running a Windows 10 64-bit operating system (Microsoft Corporation, Albuquerque, USA).

## 3. Results and Discussion

### 3.1. Frequency-Dependent DPs

Figure 1 displays the semilog plot of ε′ and ε″ as functions of the frequency at different moisture contents and temperatures. At moisture contents of 10 and 30% wb, both ε′ and ε″ displayed a nonlinear decrease with increasing frequency characterized by a rapid decrease in the frequency band (10 to 300 MHz), followed by a slow decrease in the intermediate and high frequency band (300 to 3000 MHz), and this nonlinear decrease was more pronounced for increased temperatures and moisture contents. Ionic conduction is considered to be the predominant polarization mechanism at low frequencies, and the strengths gradually diminish with increasing frequency [22]. At a constant frequency, a high temperature resulted in high DP values, whereas at a constant temperature and given frequency, a high moisture content caused high DP values (Figure 1). This behavior can be explained by the fact that a high temperature and moisture content improve the ionic mobility and dipole rotation, which result in increased DP values [30]. However, at a low moisture content of 10%, all the ε′ and ε″ values of the pecan kernels were less than 8 at any measured temperature. Similar DP values were also observed by Wang et al. [31] for walnut kernels at moisture contents of 7.5% wb, from 1 to 1800 MHz. This result may be attributed to the high fat contents of pecan kernels, as shown in Table 1.

### 3.2. Moisture- and Temperature-Dependent DPs

The 3D plots of DPs at 27, 40, 915, and 2450 MHz are presented for pecan kernels with a moisture content range of 10−30% wb and a temperature range of 5–65 °C in Figure 2 and Figure 3. Overall, both the ε′ and ε′’ values of the pecan kernel samples in the RF range were significantly larger than those in the MW frequency range. At a certain frequency, an increase in the temperature and moisture content resulted in significant increases in ε′ and ε″, although the values increased more in the high temperature and moisture content range. For instance, at 27 MHz, when the moisture content increased from 10% to 30% wb, ε′ increased from 3.08 to 5.51 at 5 °C, from 6.67 to 18.01 at 45 °C, and from 8.09 to 20.39 at 65 °C. For the same moisture content and frequency, ε″ increased from 0.37 to 1.05 at 5 °C, from 5.15 to 13.06 at 45 °C, and from 7.85 to 25.86 at 65 °C. Compared with the values of the DPs in the RF range, the DPs at 980 and 2450 MHz (MW frequency) had much lower values for the same moisture and temperature levels. The same tendency has been observed at a given frequency in pistachio and peanut kernels [23,24], where the DPs at a high moisture content and temperature had significantly larger values than those at low moisture and temperature.

The pecan kernel samples included a mixture of constituents with different dielectric behaviors and DPs. Due to the polar nature and solvent effect of water molecules, the DP values (ε′, 80.4) were much larger than those of other matter; thus, the water molecule was the main dipole that was most responsible for the dielectric heating of materials. The water in the nut samples was mostly present in the form of free and bound water, in which the dipole polarization aroused by free water was considerably larger than that caused by bound water. In low-moisture nut kernels, most water molecules exist in the state of bound water in combination with proteins or carbohydrates [23]. Therefore, ε′ and ε″ were very low, irrespective of the temperature and frequency. As the moisture content increased from 10% to 30% wb in the sample, the amount of free water, the ionic conduction, and the bulk density increased. All of these parameters contributed to the increase in the values of the DPs [32]. As displayed in Figure 3, the ε′ and ε″ values of the pecan kernels in all RF and MW bands increased more remarkably with an increase in the moisture content at 65 °C than at 25 °C. The increasing temperature caused an increase in the thermal motion of the molecules and a decrease in the viscosity of the heated material [17]. Thus, the ionic conductivity increased.

In a batch RF or MW drying process of pecan kernels, the increase in ε″ with an increased moisture content may lead to a potentially beneficial phenomenon, which is commonly referred to as the “moisture leveling effect” [30]. The high-moisture areas in the pecan samples could absorb increased electromagnetic energy and be heated preferentially. Thus, a faster heating rate and higher temperature would result in the high-moisture areas than in the low-moisture areas. Consequently, more water vaporization occurred in the high-moisture areas than in other areas, which finally led to a relatively uniform moisture content in the dried product. Conversely, the increase in ε″ with increasing temperature may lead to a situation called “thermal runaway” when drying through dielectric heating [22,33]. Areas with high temperature have a large ε″ value, which results in dielectric heating, which in turn causes more hot spots. This run-away phenomenon may lead to an increased vapor pressure gradient that promotes water migration from the inner part of the kernel to the surface of the sample. However, a very high temperature would adversely affect product quality. To avoid a very high temperature in dielectric heating, effective approaches should be taken to maintain a balance between the electromagnetic field input and energy output, such as surface air flowing in association with internal dielectric heating.

### 3.3. Regression Models for the DPs of Pecans

The polynomial regression models simulating the relationship between DPs of the pecan nuts and the temperature and moisture contents at frequencies of 27, 40, 915, and 2450 MHz are presented in Table 3 and Table 4. Quadratic polynomial regression equations are the most suitable option for associating DPs with the temperature and moisture content. An analysis of variance was performed to test whether the temperature and moisture content had significant influences on the polynomial regression models (Table 5 and Table 6). The linear term and interaction terms of Moisture content (M) and Temperature (T) had significant effects on the models (*p* < 0.05). All the models exhibited a good fit with the data at a significance level of *p* < 0.0001, with a coefficient of determination (R) higher than 0.9819. These results indicate that the polynomial models could be used to precisely predict the ε′ and ε″ values of the pecan kernels in a known moisture content range of 10–30% wb, a temperature range of 5–65 °C, and four specific frequencies.

### 3.4. Effect of the Salt Levels on DPs

The DPs of the pecan kernels with different salt levels were measured at frequencies of 27, 40, 915, and 2450 MHz in the temperature range from 5 to 65 °C, as listed in Table 7. For a certain temperature, ε′ did not exhibit an obvious change with increasing salt concentration, whereas ε″ increased significantly with increasing salt level, especially in the RF band. The nonsignificant effect of salt on ε′ is in accordance with the observations presented by other researchers for various food materials [24,34,35]. The nonsignificant effect may be because the added salts can decrease the water activity and diminish the polarization characteristics [25]. In this research, the addition of an electrolyte (NaCl) did not considerably affect ε′; however, the addition did have a marked effect on ε″. The addition of salt to the material could trigger large-scale electrophoretic migration when placed in an electric field, which would promote an increase in ε″ by ionic conduction. The dielectric response of salts is closely connected with the effective nuclear charge and depends on the volume and charge of dissolved salt ions [25]. Ionic conduction mainly occurs in the RF range, and dipole rotation is the main functional mechanism of dielectric heating in the MW range [36]. The addition of salt addition provides an increased contribution to the development of ionic conduction in the RF range; thus, the increase in ε″ was higher at 27 and 40 MHz than at 915 and 2450 MHz; for instance, when the salt concentration increased from non-salted to strongly salted at 25 °C, ε″ increased by 627% at 27 MHz and 390% at 915 MHz. Furthermore, significant increases in ε′ and ε″ were observed over the temperature range of 5 to 65 °C for both the salt-enriched and non-salted samples at four frequencies. For samples with salt, the ionic loss from electrophoretic migration increased with temperature. The variation of DPs in terms of the temperature and frequency was similar for both the salted and non-salted pecan kernel samples. The addition of salt significantly increased the values of ε″, which indicates the need to develop a separate drying scheme for salted and non-salted pecan nuts.

### 3.5. Penetration Depth

The power dp computed from the measured ε′ and ε″ of the pecan kernels at four frequencies, four temperatures, five moisture contents, and three salt levels is listed in Table 8 and Table 9. The dp in the RF range was considerably higher than that in the MW range, and the dp decreased with increasing temperature and moisture content. For instance, the dp at 27 MHz was distributed from 1339.26 to 31.89 cm for the pecan kernels depending on the kernel moisture content and temperature, whereas the dp at 915 MHz was distributed from 47.29 to 7.15 cm. As the moisture content of the pecan samples increased from 10% to 30% wb, the dp decreased from 707.56 to 94.3 cm at 45 °C and 27 MHz. As the temperature rose from 5 to 65 °C, the dp decreased from 382.88 to 58.94 cm at 40 MHz and a moisture content of 20%. For salt-added pecan samples with a moisture content of 15%, the dp decreased with increasing salt level. These results agree with those of previous reports on peanuts [23] and pistachio kernels [24].

The relatively shallow penetration depth at MW frequencies indicated that MW energy only penetrates into the shallow layers of pecan kernels, which results in considerable surface heating. The high penetration depths in the RF range led to a relatively uniform distribution of the electromagnetic field in the bulk mass of pecan kernels. Thus, the uniformity of heating was improved, which is one of the most important advantages of RF heating compared with MW drying [37]. The actual stacked thickness in a batch drying process of pecan nuts should be lower than 27 and 5 cm at 27 and 2450 MHz, respectively.

### 3.6. Comparison of Simulated and Experimental Heating Rates of Pecan Kernels

Figure 4 shows a comparison between the experimental and simulated temperature-time profiles of pecan kernels with five moisture contents of 10%, 15%, 20%, 25%, and 30% during 5 min RF heating with an electrode gap of 150 mm. It can be seen for all the tested samples that the simulation results using measured DP values accorded with experimental test temperature profiles at five moisture levels. Within the 5 min RF heating period, all heating curves were relatively linear, and the heating rates of pecan kernels increased significantly (*p* < 0.05) with increasing moisture content from 10% to 25%. However, the heating rate then decreased as the concentration increased from 25% to 30%. The 25% sample had the highest heating rate (11.2 °C /min). Similar trends were also found by Zhang et al. [23] for the heating rate of peanut kernels, in which the increase in moisture content resulted in increasing ε″, which in turn caused an initial increase and then a decrease in RF heating rates. Jiao et al. [38] reported that the maximum heating rate was reached when the values of ε′ and ε″ were close to one another.

## 4. Conclusions

The DP values of pecan kernels decreased with increasing frequency and increased with increasing water content and temperature. The ε′ and ε″ values of the pecan kernels decreased substantially with increasing frequency in the measured RF range, whereas the values decreased gradually in the measured MW range. The addition of salt led to marginal increases in ε′ but sharp increases in ε″, especially in the RF range. At four specific frequencies, the relationship between the DPs and the temperature and moisture content could be quantitatively described with a battery of quadratic polynomial equations, which could be adopted to precisely predict the ε′ and ε″ values from the temperature and moisture content. The penetration depths of electromagnetic power in the pecan kernel decreased sharply with increasing frequency, temperature, and moisture content. Consequently, RF dielectric heating could provide relatively more uniform heating. Thus, RF dielectric heating is especially suited for large-scale treatments.

## Figures and Tables

**Figure 1 foods-08-00385-f001:**
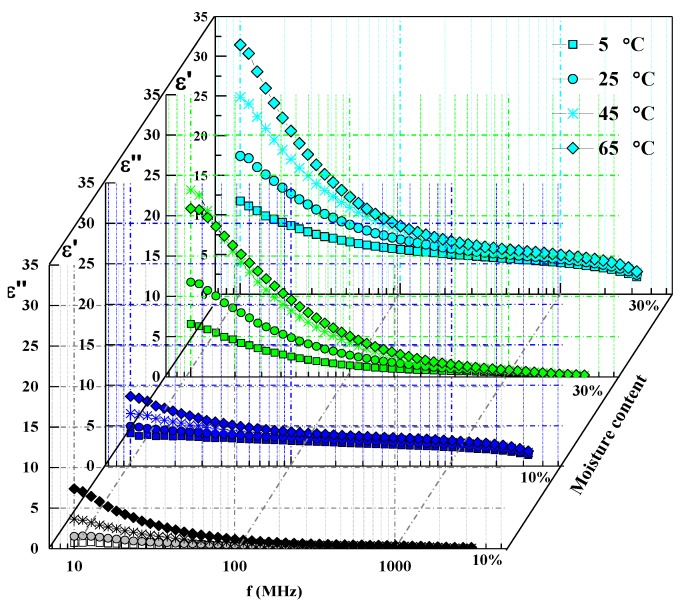
Frequency-dependent dielectric constant (ε’) and loss factor (ε’’) of the pecan kernels at four temperatures and moisture contents of 10% and 30%.

**Figure 2 foods-08-00385-f002:**
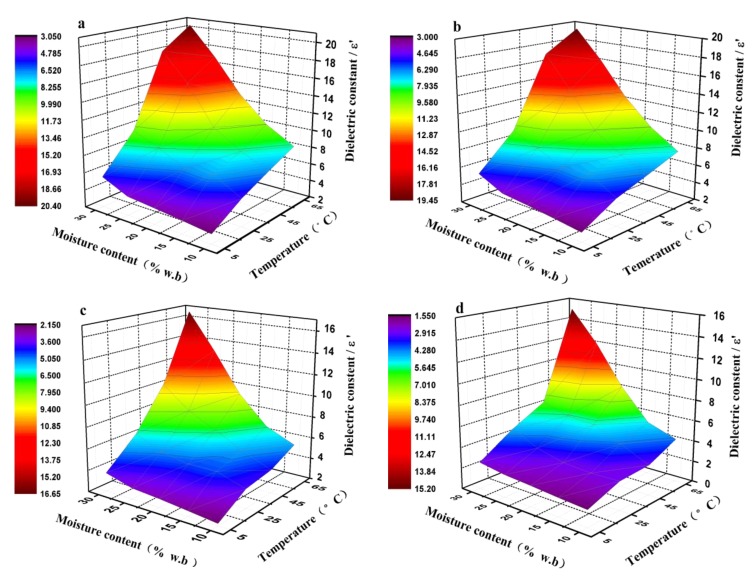
Three-dimensional representation of the dielectric constants of the pecan kernel as functions of the moisture content and temperature at frequencies of (**a**) 27, (**b**) 40, (**c**) 915, and (**d**) 2450 MHz.

**Figure 3 foods-08-00385-f003:**
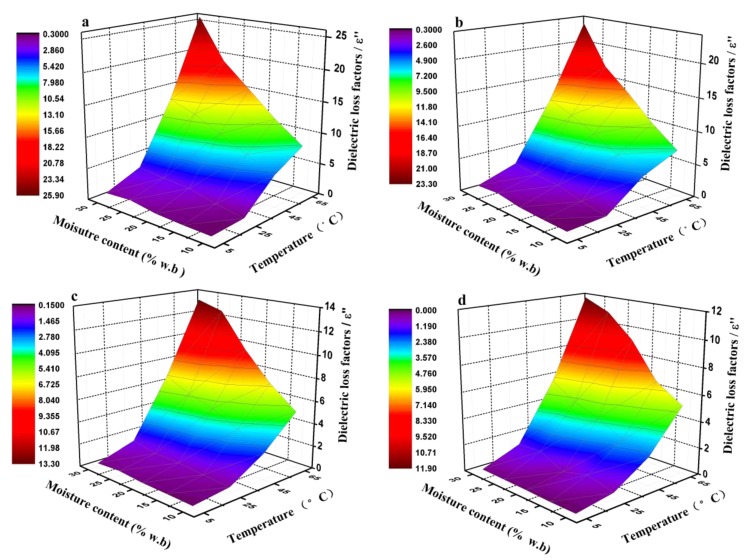
Dielectric loss factors of the pecan kernel samples as functions of the moisture content and temperature at frequencies of (**a**) 27, (**b**) 40, (**c**) 915, and (**d**) 2450 MHz over a moisture content range of 10–30% wb and a temperature range of 5–65 °C.

**Figure 4 foods-08-00385-f004:**
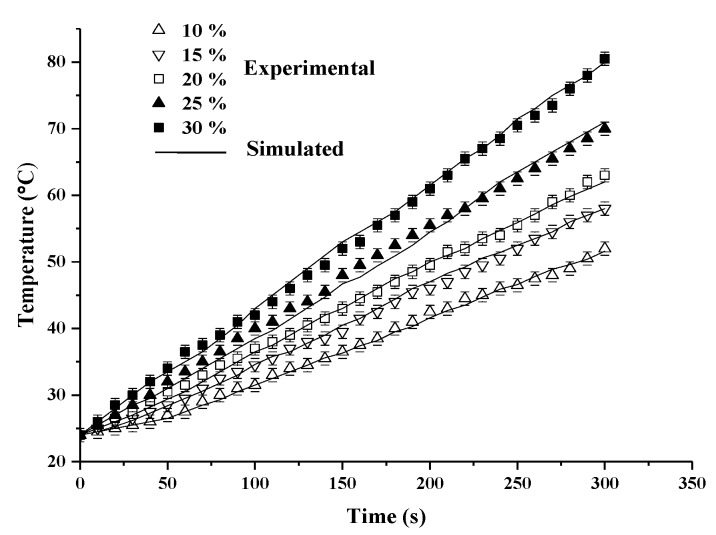
Experimental and simulated temperature-time histories of pecan kernels with a moisture content from 10% to 30% wb when subjected to RF heating for 5 min with an electrode gap of 150 mm.

**Table 1 foods-08-00385-t001:** Chemical compositions (g/100 g, average ± SD of three replicates) of the pecan kernels.

Composition	Content	Method
Fat	67.12 ± 0.37	AOAC 948.22
Protein ^a^	14.60 ± 0.29	AOAC 950.48
Moisture	2.76 ± 0.55	AOAC 925.40
Ash	2.84 ± 0.32	AOAC 950.49
Dietary fiber	7.59 ± 0.63	AOAC 985.29
Carbohydrate	10.18 ± 0.24	Estimated by difference ^b^

^a^ Protein was calculated by considering a nitrogen conversion factor of 5.3. ^b^ Carbohydrate content = 100% − (% moisture + % protein + % fat + % ash + % dietary fiber).

**Table 2 foods-08-00385-t002:** The densities and specific heat of pecan kernel at five moisture contents.

Moisture Content (% wb)	Density ± SD (g cm^−3^)	Specific Heat (J kg^−1^ °C^−1^)
10	0.8150 ± 0.0102	970 ± 35
15	0.9431 ± 0.0075	1138 ± 46
20	1.0005 ± 0.0037	1267 ± 53
25	1.1024 ± 0.0008	1382 ± 37
30	1.2715 ± 0.0103	1469 ± 48

**Table 3 foods-08-00385-t003:** Regression equations for the dielectric constants of the pecan kernels as functions of the moisture content (10% ≤ M ≤ 30%) and temperature (5 °C ≤ T ≤ 65 °C) at specific frequencies.

Frequency (MHz)	Dielectric Constant (*ε′*)
27	*ε*′ = 5.165 − 0.325*M* + 0.022*T* + 0.009*M*^2^ − 0.001*T*^2^ + 0.01*M*·*T* (2)
40	*ε*′ = 5.254 − 0.321*M* + 0.007*T* + 0.008*M*^2^ − 0.001*T*^2^ + 0.009*M*·*T* (3)
915	*ε*′ = 5.615 − 0.301*M* − 0.107*T* + 0.007*M*^2^ + 0.001*T*^2^ + 0.009*M*·*T* (4)
2450	*ε*′ = 5.151 − 0.255*M* − 0.178*T* + 0.005*M*^2^ + 0.002*T*^2^ + 0.008*M*·*T* (5)

**Table 4 foods-08-00385-t004:** Regression equations for the dielectric loss factors of the pecan kernels as functions of the moisture content (10% ≤ M ≤ 30%) and temperature (5 °C ≤ T ≤ 65 °C) at specific frequencies.

Frequency (MHz)	Dielectric Loss Factor (*ε″*)
27	*ε*″ = 3.649 − 0.245*M* − 0.238*T* + 0.003*M*^2^ + 0.003*T*^2^ + 0.014*M*·*T* (6)
40	*ε*′ = 3.318 − 0. 215*M* − 0.218*T* + 0.003*M*^2^ − 0.003*T*^2^ + 0.013*M*·*T* (7)
915	*ε*″ = 1.153 + 0.004*M* − 0.177*T* − 0.002*M*^2^ + 0.002*T*^2^ + 0.007*M*·*T* (8)
2450	*ε*″ = 1.125 − 0.031*M* − 0.153*T* − 0.001*M*^2^ + 0.002*T*^2^ + 0.006*M*·*T* (9)

**Table 5 foods-08-00385-t005:** Significance of the probability of the regressed models in Equations (2)–(5) for the pecan kernel samples at four specific frequencies.

Variance and R	27 MHz (2)	40 MHz (3)	915 MHz (4)	2450 MHz (5)
M	<0.0001	<0.0001	<0.0001	<0.0001
T	<0.0001	<0.0001	<0.0001	<0.0001
M^2^	0.1560	0.0523	0.0579	0.3928
T^2^	0.0654	0.2180	0.0904	0.0062
M × T	0.0460	<0.0001	0.0065	0.0062
Model	<0.0001	<0.0001	<0.0001	<0.0001
*R*	0.9787	0.9801	0.9752	0.9419

M, Moisture content; T, Temperature.

**Table 6 foods-08-00385-t006:** Significance of the regressed models in Equations (6)–(9) for the pecan kernel samples at four specific frequencies.

Variance and R	27 MHz (6)	40 MHz (7)	915 MHz (8)	2450 MHz (9)
M	<0.0001	0.0029	<0.0001	<0.0001
T	<0.0001	0.0009	<0.0001	<0.0001
M^2^	0.0005	0.0523	<0.0001	<0.0001
T^2^	0.6775	0.7083	0.5121	0.7620
M × T	<0.0001	<0.0001	<0.0001	0.0002
Model	<0.0001	<0.0001	<0.0001	<0.0001
*R*	0.9753	0.9730	0.9850	0.9897

**Table 7 foods-08-00385-t007:** Dielectric properties of pecan nuts with different salt contents (moisture content: 15%).

Samples	T (°C)	Dielectric Properties	Frequency (MHz) 2450 MHz			
			27	40	915	2450
No salt	5	ε′ ± SD	3.47 ± 0.03	3.38 ± 0.04	2.50 ± 0.05	1.77 ± 0.01
		ε″ ± SD	0.47 ± 0.01	0.44 ± 0.02	0.24 ± 0.01	0.11 ± 0.00
	25	ε′ ± SD	8.97 ± 0.04	7.73 ± 0.06	4.66 ± 0.03	2.65 ± 0.07
		ε″ ± SD	3.35 ± 0.03	3.20 ± 0.01	1.71 ± 0.04	1.01 ± 0.02
	45	ε′ ± SD	15.09 ± 0.08	17.53 ± 0.09	6.33 ± 0.06	4.18 ± 0.05
		ε″ ± SD	10.18 ± 0.04	8.58 ± 0.05	3.69 ± 0.06	2.78 ± 0.03
	65	ε′ ± SD	20.01 ± 0.11	19.56 ± 0.09	8.57 ± 0.12	6.19 ± 0.08
		ε″ ± SD	15.19 ± 0.13	10.03 ± 0.14	6.96 ± 0.07	6.49 ± 0.10
Light- salt	5	ε′ ± SD	5.32 ± 0.09	5.03 ± 0.07	3.82 ± 0.05	3.15 ± 0.04
		ε″ ± SD	1.28 ± 0.03	1.19 ± 0.06	0.86 ± 0.05	0.73 ± 0.04
	25	ε′ ± SD	11.46 ± 0.13	10.07 ± 0.14	6.87 ± 0.08	5.33 ± 0.07
		ε″ ± SD	8.58 ± 0.05	7.34 ± 0.03	2.89 ± 0.02	2.55 ± 0.05
	45	ε′ ± SD	18.93 ± 0.15	17.89 ± 0.17	8.35 ± 0.14	7.49 ± 0.13
		ε″ ± SD	15.23 ± 0.07	14.89 ± 0.07	5.96 ± 0.04	4.64 ± 0.05
	65	ε′ ± SD	23.71 ± 0.27	22.15 ± 0.32	13.84 ± 0.18	11.15 ± 0.19
		ε″ ± SD	27.02 ± 0.11	26.85 ± 0.08	9.27 ± 0.05	9.12 ± 0.04
Medium-salt	5	ε′ ± SD	7.45 ± 0.14	7.23 ± 0.16	3.96 ± 0.13	3.66 ± 0.17
		ε″ ± SD	5.18 ± 0.06	5.04 ± 0.03	3.14 ± 0.05	3.02 ± 0.02
	25	ε′ ± SD	13.97 ± 0.14	13.12 ± 0.15	7.94 ± 0.11	7.59 ± 0.07
		ε″ ± SD	14.96 ± 0.03	13.82 ± 0.06	5.15 ± 0.05	4.89 ± 0.04
	45	ε′ ± SD	20.79 ± 0.28	20.24 ± 0.35	10.64 ± 0.17	9.08 ± 0.13
		ε″ ± SD	25.89 ± 0.09	22.33 ± 0.06	7.28 ± 0.03	7.04 ± 0.04
	65	ε′ ± SD	27.43 ± 0.36	26.56 ± 0.25	14.81 ± 0.22	14.17 ± 0.18
		ε″ ± SD	34.83 ± 0.12	31.21 ± 0.09	13.14 ± 0.08	12.85 ± 0.07
Heavy-salt	5	ε′ ± SD	10.66 ± 0.12	10.14 ± 0.09	4.23 ± 0.04	3.97 ± 0.02
		ε″ ± SD	13.38 ± 0.06	11.15 ± 0.07	5.42 ± 0.05	4.25 ± 0.02
	25	ε′ ± SD	15.48 ± 0.15	14.06 ± 0.11	8.39 ± 0.09	8.01 ± 0.08
		ε″ ± SD	24.26 ± 0.06	22.09 ± 0.06	8.29 ± 0.07	7.03 ± 0.05
	45	ε′ ± SD	23.54 ± 0.23	22.78 ± 0.25	13.26 ± 0.17	12.87 ± 0.13
		ε″ ± SD	33.17 ± 0.11	28.89 ± 0.13	11.32 ± 0.08	10.13 ± 0.10
	65	ε′ ± SD	29.37 ± 0.24	28.93 ± 0.27	15.78 ± 0.18	15.32 ± 0.16
		ε″ ± SD	47.96 ± 0.13	43.37 ± 0.08	15.41 ± 0.10	14.91 ± 0.07

**Table 8 foods-08-00385-t008:** Electromagnetic energy penetration depth for pecan nuts with different moisture contents.

T (°C)	M (%)	Penetration Depth (cm)			
		27 MHz	40 MHz	915 MHz	2450 MHz
5	10	1339.26 ± 21.43	856.58 ± 18.86	47.29 ± 2.38	24.23 ± 1.37
	15	1021.34 ± 26.79	496.43 ± 13.76	34.3 ± 1.54	19.92 ± 1.29
	20	712.56 ± 20.16	382.88 ± 9.43	26.23 ± 1.17	16.31 ± 1.07
	25	397.75 ± 17.89	220.19 ± 11.87	19.21 ± 1.14	13.68 ± 0.86
	30	272.74 ± 8.89	206.36 ± 13.69	15.83 ± 0.86	10.05 ± 0.65
25	10	949.54 ± 24.71	758.12 ± 17.89	39.83 ± 4.35	22.24 ± 2.14
	15	607.56 ± 19.59	449.77 ± 14.36	26.53 ± 2.07	16.93 ± 1.69
	20	357.19 ± 11.37	319.49 ± 12.76	21.42 ± 2.49	12.50 ± 1.32
	25	287.35 ± 14.27	263.18 ± 8.63	17.75 ± 1.77	9.15 ± 0.68
	30	169.94 ± 9.25	117.19 ± 6.52	13.66 ± 1.94	8.62 ± 0.63
45	10	707.56 ± 19.74	675.33 ± 15.47	32.69 ± 2.05	16.93 ± 1.36
	15	424.49 ± 16.63	407.8 ± 14.33	22.99 ± 1.65	15.68 ± 1.17
	20	207.93 ± 15.39	177.79 ± 7.12	17.73 ± 1.44	12.49 ± 0.95
	25	153.88 ± 11.47	106.34 ± 6.26	11.96 ± 0.68	8.24 ± 0.69
	30	94.3 ± 4.48	78.89 ± 2.78	13.84 ± 1.23	6.61 ± 0.52
65	10	249.42 ± 13.86	177.45 ± 9.93	21.87 ± 1.43	14.77 ± 0.78
	15	129.86 ± 6.57	96.28 ± 7.26	16.22 ± 1.27	12.29 ± 0.63
	20	84.79 ± 7.32	58.94 ± 4.37	12.58 ± 1.83	10.02 ± 0.59
	25	55.34 ± 3.98	41.03 ± 2.55	9.94 ± 0.87	7.59 ± 0.48
	30	31.89 ± 2.17	27.48 ± 1.18	7.15 ± 0.69	5.43 ± 0.28

**Table 9 foods-08-00385-t009:** Electromagnetic energy penetration depth for pecan nuts with different salt contents.

Salt Sample	T (°C)	Penetration Depth (cm)			
		27 MHz	40 MHz	915 MHz	2450 MHz
Light	5	128.57 ± 4.54	101.75 ± 3.69	26.39 ± 1.89	18.17 ± 1.54
	25	69.65 ± 3.87	42.79 ± 2.28	19.54 ± 1.23	12.28 ± 1.17
	45	48.43 ± 3.25	31.65 ± 1.55	11.39 ± 1.64	8.64 ± 1.18
	65	29.54 ± 1.87	20.35 ± 0.98	8.53 ± 1.09	6.32 ± 0.94
Medium	5	94.79 ± 3.65	66.79 ± 2.79	22.45 ± 1.25	15.26 ± 0.63
	25	51.69 ± 2.36	38.54 ± 1.78	15.17 ± 1.07	10.19 ± 0.73
	45	37.23 ± 1.65	21.97 ± 0.84	9.31 ± 0.89	7.26 ± 0.89
	65	22.79 ± 1.23	16.29 ± 1.02	7.37 ± 0.67	6.11 ± 0.94
Heavy	5	63.88 ± 2.74	40.26 ± 1.59	18.46 ± 1.05	11.37 ± 0.44
	25	42.69 ± 2.29	27.83 ± 1.34	13.65 ± 1.17	8.29 ± 0.72
	45	29.43 ± 1.69	18.48 ± 1.27	8.14 ± 1.19	6.39 ± 0.49
	65	13.76 ± 0.88	11.28 ± 1.19	7.02 ± 0.65	4.21 ± 0.83

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
