# Peer review of "Effects of Moisture, Temperature, and Salt Content on the Dielectric Properties of Pecan Kernels during Microwave and Radio Frequency Drying Processes"

_foods, 2019, doi:10.3390/foods8090385_

Round 1

Reviewer 1 Report

This paper reports the measured dielectric properties of pecan kernels at 27, 40, 915, and 2450 MHz z as functions of moisture content and temperature at four different salt contents.  The work is well described, and the paper is reasonably well written, but the writing could be improved with a careful editing by someone accustomed to technical writing in English.  Specific corrections that I noted in reading through the manuscript, which would improve the quality are as follows:

Line 13 (first line of the Abstract): ”Dielectric properties of electromagnetic fields” makes no sense, so the sentence should be revised as “Dielectric properties of materials influence the interaction of electromagnetic fields with those materials and are therefore important in designing effective dielectric heating processes.”

Line 40: Reference no. 9 cited is work with almonds, and it should not be cited for pecans.  “Pecans kernels” should be replaced with “Pecan kernels”

Line 43:  Replace “pecan” with “pecans”   Similar problems with plural forms in the manuscript should be corrected by editing for proper English.

Line 57:  In the equation for the relative complex permittivity, the + sign should be replaced with a – sign, which is the common accepted practice in dielectrics literature.

Line 60: For accuracy, “The higher the DP values” should be replaced with “the higher the dielectric loss factor value”

Line 81:  The term “real density” is inappropriate for “kernel density” because both “kernel density” and “bulk density” are real densities.

Lines 81 and 82:  This is a false statement, which should be removed, because the dielectric properties of pecan kernels have been reported previously as influenced by moisture content and temperature.  The authors are apparently unfamiliar with similar measurements reported earlier as follows:

Frequency and moisture dependence of the dielectric properties of chopped pecans.  Stuart O. Nelson.  Transactions of the ASAE 24(6):1573-1576, 1981.  (Dielectric constants and dielectric loss factors of chopped pecan nut-meat samples are presented graphically as mean values for six cultivars, along with bulk and kernel density data, over a moisture-content range from 3% to 9% and over the frequency range from 50 kHz to 10 GHz.)

RF dielectric heating for pecan weevil control.  S. O. Nelson and J. A. Payne.  Transactions of the ASAE 25(2):456-458, 464, 1982.  (Description of 40-MHz exposures of infested in-shell pecans and weevil larvae in broken pecan pieces, resulting weevil mortality data, and effects of exposures on pecan germination)

Effects of dielectric and steam heating treatments on the pre-storage and storage color characteristics of pecan kernels.  S. D. Senter, W. R. Forbus, Jr., S. O. Nelson, and R. J. Horvat.  Journal of Food Science 49(6):1532-1534, 1984.  (Steam heating caused a darkening of the pecan kernels that was accentuated during storage, whereas dielectric heating did not produce darkening initially or during storage)

Temperature dependence of the dielectric properties of pecans.  K. C. Lawrence, S. O. Nelson, and A. W. Kraszewski.  Transactions of the ASAE 35(l):251-255, 1992. (Data presented graphically over 0 to 40 °C temperature range at frequencies from 0.1 to 110 MHz and moisture contents from 3%  to 9%, wet basis)

Temperature dependence of the dielectric properties of pecans.  K. C. Lawrence, S. O. Nelson, and A. W. Kraszewski.  Transactions of the ASAE 35(l):251-255, 1992. (Data presented graphically over 0 to 40 °C temperature range at frequencies from 0.1 to 110 MHz and moisture contents from 3%  to 9%, wet basis)

Dielectric Properties of Agricultural Materials and Their Applications. Nelson, S. O.  Elsevier - Academic Press, Amsterdam, Boston, Heidelberg, London, New York, 2015.

All of these references include information on dielectric properties of pecans or RF dielectric heating of pecans, and several of them should be included in the review of literature.

Line 85:  Replace “with four temperature scopes.” with “at four temperature levels.”

Line 125: Replace “sheets” with “discs”, which is a much better term for describing the compressed samples.  Dimensions of the samples should also be given.

Line 143 (Table 2 title): Replace “Real densities” with “Kernel densities” of “Kernel sample densities”

Line 178:  Replace “Voltage on the upper electrode” with “Voltage between the electrodes”

Line 187: Replace “meshing create and optimize” with “mesh creation and optimization”

                Replace “seting” with “setting”

Line 211: Explain what “This” refers to explicitly.

Line 228 and 229:  Delete “a large decrease was observed in” and insert “much lower values” between “frequency),” and “for”

Line 254:  Replace “were observed” with “would result”

Line 306:  Replace “improved” with “increased”

There are also questions related to pecan sample densities that need to be explained.  How were pecan kernel densities determined?  The dielectric loss factor that should be used in equation 2 for calculating the heating rate is not the loss factor of the kernel, but instead the loss factor of  the  kernel and air mixture when exposed in the dielectric heating system.  Also in that equation, bulk density of the pecan-air sample should be used instead of the kernel density. The dependence of kernel density and bulk density of pecan kernels on moisture content are also provided in the first (1981) reference listed above.

Author Response

Aug. 27, 2019

Dear Professor:

Thank you very much for your review on our manuscript entitled “Effect of Moisture, Temperature, and Salt Content on the Dielectric Properties of Pecan Kernels During Microwave and Radio Frequency Drying Processes.” (Foods-582370). We really appreciate the reviewers’ comments and your kind decision. According to the comments, we carefully amended the manuscript. We expect the revised manuscript will better fulfill the publication requirement of the Foods.

The point to point answers to reviewers’ comments are listed below, which will be helpful for your re-evaluation of our manuscript.

Reviewers' comments:

This paper reports the measured dielectric properties of pecan kernels at 27, 40, 915, and 2450 MHz z as functions of moisture content and temperature at four different salt contents. The work is well described, and the paper is reasonably well written, but the writing could be improved with a careful editing by someone accustomed to technical writing in English. Specific corrections that I noted in reading through the manuscript, which would improve the quality are as follows:

Line 13 (first line of the Abstract): ”Dielectric properties of electromagnetic fields” makes no sense, so the sentence should be revised as “Dielectric properties of materials influence the interaction of electromagnetic fields with those materials and are therefore important in designing effective dielectric heating processes.”

Response: Thanks for your kindly reminding. We are sorry for our incomplete description on the dielectric properties. In the revised manuscript, we have carefully revised the sentence according to your kindly suggestion. (page 1, line 13).

Line 40: Reference no. 9 cited is work with almonds, and it should not be cited for pecans. “Pecans kernels” should be replaced with “Pecan kernels”

Response: We are sorry for our mistakes, and we have deleted the Reference no. 9 cited (page 1, lines 42) in the revised manuscript.

Line 43: Replace “pecan” with “pecans” Similar problems with plural forms in the manuscript should be corrected by editing for proper English.

Response: We are sorry for our mistakes, we have carefully revised the imilar problems with plural forms in the revised manuscript: page 1, line 2 and line43;page 2, line 45, 67, 73, 74, 85;page 3, 133;page 5,line211;page 12, line347.

Line 57: In the equation for the relative complex permittivity, the + sign should be replaced with a – sign, which is the common accepted practice in dielectrics literature.

Response: We are sorry for our mistakes, thank you for your kindly suggestion, we have revised the manuscript according to the comments in page 2, line 59.

Line 60: For accuracy, “The higher the DP values” should be replaced with “the higher the dielectric loss factor value”

Response: Thanks for your kindly reminding. We have carefully revised the manuscript according to your kindly suggestion

Line 81: The term “real density” is inappropriate for “kernel density” because both “kernel density” and “bulk density” are real densities.

Response: We are sorry for our inappropriate description on density. In the revised manuscript, the “real density” have be replaced with “kernel bulk density” (page 2, line 84).

Lines 81 and 82: This is a false statement, which should be removed, because the dielectric properties of pecan kernels have been reported previously as influenced by moisture content and temperature. The authors are apparently unfamiliar with similar measurements reported earlier as follows:

Frequency and moisture dependence of the dielectric properties of chopped pecans. Stuart O. Nelson. Transactions of the ASAE 24(6):1573-1576, 1981. (Dielectric constants and dielectric loss factors of chopped pecan nut-meat samples are presented graphically as mean values for six cultivars, along with bulk and kernel density data, over a moisture-content range from 3% to 9% and over the frequency range from 50 kHz to 10 GHz.)

RF dielectric heating for pecan weevil control. S. O. Nelson and J. A. Payne. Transactions of the ASAE 25(2):456-458, 464, 1982. (Description of 40-MHz exposures of infested in-shell pecans and weevil larvae in broken pecan pieces, resulting weevil mortality data, and effects of exposures on pecan germination)

Effects of dielectric and steam heating treatments on the pre-storage and storage color characteristics of pecan kernels. S. D. Senter, W. R. Forbus, Jr., S. O. Nelson, and R. J. Horvat. Journal of Food Science 49(6):1532-1534, 1984. (Steam heating caused a darkening of the pecan kernels that was accentuated during storage, whereas dielectric heating did not produce darkening initially or during storage)

Temperature dependence of the dielectric properties of pecans. K. C. Lawrence, S. O. Nelson, and A. W. Kraszewski. Transactions of the ASAE 35(l):251-255, 1992. (Data presented graphically over 0 to 40 °C temperature range at frequencies from 0.1 to 110 MHz and moisture contents from 3% to 9%, wet basis)

Temperature dependence of the dielectric properties of pecans. K. C. Lawrence, S. O. Nelson, and A. W. Kraszewski. Transactions of the ASAE 35(l):251-255, 1992. (Data presented graphically over 0 to 40 °C temperature range at frequencies from 0.1 to 110 MHz and moisture contents from 3% to 9%, wet basis)

Dielectric Properties of Agricultural Materials and Their Applications. Nelson, S. O. Elsevier - Academic Press, Amsterdam, Boston, Heidelberg, London, New York, 2015.

All of these references include information on dielectric properties of pecans or RF dielectric heating of pecans, and several of them should be included in the review of literature.

Response: We are grateful for reviewers’ kindly suggestion and so great help !  We are sorry for our incorrect description and insufficient understanding of background knowledge about the dielectric properties of pecan kernels. We have deleted the sentence (page 2,line 85-86) and carefully read these references you suggested. We have choose the refence entitled “RF dielectric heating for pecan weevil control. S. O. Nelson and J. A. Payne. Transactions of the ASAE 25(2):456-458, 464, 1982” and “Effects of dielectric and steam heating treatments on the pre-storage and storage color characteristics of pecan kernels” in the review of literature(page 2, line 77-79).

Line 85: Replace “with four temperature scopes.” with “at four temperature levels.”

Response: We are sorry for our mistakes and we have carefully revised the

manuscript according to your kindly suggestion (page2, line 90-91)

Line 125: Replace “sheets” with “discs”, which is a much better term for describing the compressed samples. Dimensions of the samples should also be given.

Response: Thanks for your kindly reminding , and we have carefully revised the

manuscript according to your kindly suggestion.

Line 143 (Table 2 title): Replace “Real densities” with “Kernel densities” of “Kernel sample densities”

Response: Thanks for your kindly reminding , and we have carefully revised the

manuscript according to your kindly suggestion.

Line 178: Replace “Voltage on the upper electrode” with “Voltage between the electrodes”

Response: Thanks for your kindly reminding , and we have carefully revised the

manuscript according to your kindly suggestion.

Line 187: Replace “meshing create and optimize” with “mesh creation and optimization”

Replace “seting” with “setting”

Response:

Thank you for your insightful suggestion, and we have carefully revised the manuscript according to your kindly suggestion.

Line 211: Explain what “This” refers to explicitly.

Response: Thank you for your insightful question. We are sorry for our unclear expression. This (page 6, line 213 ) refer to the result of the previous sentence “whereas at a constant temperature and given frequency, a high moisture content caused high DP values”. In order to explicitly express ,we have revised the sentence in manuscript.

Line 228 and 229: Delete “a large decrease was observed in” and insert “much lower values” between “frequency),” and “for”

Response: Thanks for your kindly reminding , and we have carefully revised the manuscript according to your kindly suggestion.

Line 254: Replace “were observed” with “would result”

Response:

Thanks for your kindly reminding , and we have carefully revised the manuscript according to your kindly suggestion.

Line 306: Replace “improved” with “increased”

Response:

Thanks for your kindly reminding, and we have carefully revised the manuscript according to your kindly suggestion.

There are also questions related to pecan sample densities that need to be explained. How were pecan kernel densities determined? The dielectric loss factor that should be used in equation 2 for calculating the heating rate is not the loss factor of the kernel, but instead the loss factor of the kernel and air mixture when exposed in the dielectric heating system. Also in that equation, bulk density of the pecan-air sample should be used instead of the kernel density. The dependence of kernel density and bulk density of pecan kernels on moisture content are also provided in the first (1981) reference listed above.

Response: Thank you for your insightful question. The determination of pecan kernel bulk densities used the method of the liquid displacement method, in which toluene(C7H8) was used as the displaced liquid and calculate density by accurately measuring mass and volume. In the equation 2 for calculating the heating rate, the bulk density was the same as the kernel density measured previously.

Thank you for your kindly consideration, and we are looking forward to hearing

from you again.

With best regards!

Zhang jigang

Tobacco Research Institute of Chinese Academy of Agricultural Sciences

Qingdao, Shandong 266101, China

Tel: 0532-88701035

Email: zjg79@ahau.edu.cn

Reviewer 2 Report

A thorough study on dieletric properties dependence on substrate features is accomplished in this paper.

The simulation part is rather weak. The RF heating simulaion is presented as it would cover the entire frequency range, whereas it is well known that MW-specific simulations can be carried out of different grounds. The Authors are therefore kindly asked to clarify this point.

Please find minor corrections and considerations in the following list.

ABSTRACT:

temperate>temperature

INTRODUCTION

51: to induce dielectric polarization underlying dipole polarization. Clearify.

52: this is an important point and should make clear to the Reader. Heat generation depends on the distribution of free water in the substrate. A consolidated notion is that MWs mobilize free water, which during process tends to accumulate in the substrate's bottom. Likewise, heat generation (and temperature increase, somewhat mediated by latent heat evaporation) will be considerably greater at substrate bottom. Using the word "throughout" the Reader will be invited to assume that heat generation will happen more or less uniformly in the substrate volume. Please account for this notion, including pertinent literature reference.

109: Due to the inherent nonuniformity of the employed drying process (the "blast" drying oven is left undescribed), it seems necessary to add the following line to the paragraph, or a similar one: the storage time was assumed to be sufficient for the residual humidity in each kernels to completely relax and reach a uniform level throughout the kernel. Maybe enforcing this notion would not seem essential at first, due to the "grounding averaging" leading to powder for the cylinder samples, but nevertheless the Readership will be informed on the inherent nonuniformity of process and how the experiment have been conducted in a rigorous fashion.

110-118: Similar relaxing time considerations should be adopted in the mass transfer process leading to different salt concentration in the samples.

187: choosing colver and seting tolerance > choosing solver and setting tolerance. solving inbuilt > ?

190: the choice of meshing elements cannot be arbitrary. Even a minimal grid independency test must be accounted for.

Author Response

Aug. 27, 2019

Dear Professor:

Thank you very much for your review on our manuscript entitled “Effect of Moisture, Temperature, and Salt Content on the Dielectric Properties of Pecan Kernels During Microwave and Radio Frequency Drying Processes.” (Foods-582370). We really appreciate the reviewers’ comments and your kind decision. According to the comments, we carefully amended the manuscript. We expect the revised manuscript will better fulfill the publication requirement of the Foods.

The point to point answers to reviewers’ comments are listed below, which will be helpful for your re-evaluation of our manuscript.

A thorough study on dieletric properties dependence on substrate features is accomplished in this paper.

The simulation part is rather weak. The RF heating simulaion is presented as it would cover the entire frequency range, whereas it is well known that MW-specific simulations can be carried out of different grounds. The Authors are therefore kindly asked to clarify this point.

Response: Thanks for your insightful comments. The COMSOL software we used in the RF heating simulaion cover the entire frequency range, also it have simulated in special frequency. Thank you very much.  

Please find minor corrections and considerations in the following list.

ABSTRACT:

temperate>temperature

Response:  Thanks for your kindly reminding, we have modified and unified the word temperature (page1, line 21).

INTRODUCTION

51: to induce dielectric polarization underlying dipole polarization. Clearify.

Response: Thank you for your kindly question. A dielectric material is one which contains either permanent or induced dipoles which when placed between two electrodes acts as a capacitor, i.e. the material allows charge to be stored and no dc conductivity is observed between the plates. The polarisation of dielectrics arises from the finite displacement of charges or rotation of dipoles in an electric field and should not be confused with conduction which results from translational motion of the charges when the electric field is applied. At the molecular level polarisation involves either the distortion of the distribution of the electron cloud within a molecule or the physical rotation of molecular dipoles.

52: this is an important point and should make clear to the Reader. Heat generation depends on the distribution of free water in the substrate. A consolidated notion is that MWs mobilize free water, which during process tends to accumulate in the substrate's bottom. Likewise, heat generation (and temperature increase, somewhat mediated by latent heat evaporation) will be considerably greater at substrate bottom. Using the word "throughout" the Reader will be invited to assume that heat generation will happen more or less uniformly in the substrate volume. Please account for this notion, including pertinent literature reference.

Response: Thank you for your insightful suggestion. The original intention of the word "throughout" using in the sentence want to express the meaning that the heated location spreads throughout the material inside. We didn't think about the problem of uniformity. Thank you for your kindly consideration. We replace “throughout” with “inside” in the revised manuscript.

109: Due to the inherent nonuniformity of the employed drying process (the "blast" drying oven is left undescribed), it seems necessary to add the following line to the paragraph, or a similar one: the storage time was assumed to be sufficient for the residual humidity in each kernels to completely relax and reach a uniform level throughout the kernel. Maybe enforcing this notion would not seem essential at first, due to the "grounding averaging" leading to powder for the cylinder samples, but nevertheless the Readership will be informed on the inherent nonuniformity of process and how the experiment have been conducted in a rigorous fashion.

Response: Thanks for your kindly reminding and suggestion. In the revised manuscript, we added the following line to the paragraph (page 3, line 115) : The storage time was assumed to be sufficient for the residual humidity in each kernels to completely relax and reach a uniform level throughout the kernel.

110-118: Similar relaxing time considerations should be adopted in the mass transfer process leading to different salt concentration in the samples.

Response:  Thank you for your kindly suggestion, we are sorry for our negligence to this relaxing time considerations, and we have added similar related tips for relaxation time in revised manuscript. 

187: choosing colver and seting tolerance > choosing solver and setting tolerance. solving inbuilt > ?

Response: Thank you for your kindly question. We have ammeded in the revised manuscript.

190: the choice of meshing elements cannot be arbitrary. Even a minimal grid independency test must be accounted for.

Response: Thank you for your kindly comment. the choice of meshing elements includes tetrahedral,triangular,linear,vertex elements. We have amended in the revised manuscript.
